# Long-term records of glacier surface velocities in the Ötztal Alps (Austria)

Martin Stocker-Waldhuber[1/2], Andrea Fischer[1], Kay Helfricht[1], Michael Kuhn[3]

[1]Institute for Interdisciplinary Mountain Research, Austrian Academy of Sciences, Innsbruck, 6020, Austria
[2]Department of Geography, Physical Geography, Catholic University of Eichstätt-Ingolstadt, 85072, Germany
[3]Institute of Atmospheric and Cryospheric Sciences, University of Innsbruck, Innsbruck, 6020, Austria

*Correspondence to*: Martin Stocker-Waldhuber (martin.stocker-waldhuber@oeaw.ac.at)

**Abstract.** Climatic forcing affects glacier mass balance, which causes changes in ice flow dynamics and glacier length changes on different time scales. Mass balance and length changes are operationally used for glacier monitoring, whereas only a few time series of glacier dynamics have been recorded. Here we present a unique dataset of yearly-averaged ice-flow velocity measurements at stakes and stone lines covering more than 100 years on Hintereisferner and more than 50 years on Kesselwandferner. Moreover, the dataset contains subseasonal variations of ice flow from Gepatschferner and Taschachferner covering almost 10 years. The ice flow velocities on Hintereisferner and especially on Kesselwandferner show great variations between advancing and retreating periods, with magnitudes increasing from the stakes at higher elevations to the lower elevated stakes, making ice flow records at ablation stakes a very sensitive indicator of glacier state. Since the end of the latest glacier advances from the 1970s to the 1980s, the ice flow velocities have decreased continuously, a strong indicator of the negative mass balances of the glaciers in recent decades. The velocity data sets of the four glaciers are available at https://doi.pangaea.de/10.1594/PANGAEA.896741.

## 1 Introduction

The fluctuation of glaciers has become an icon of climate change, after Agassiz (1847) hypothesized the theory of ice ages, which was then confirmed by Penck and Brückner (1909) and further substantiated with isotope analysis on deep sea sediment and polar ice cores (Hays et al., 1976; Shackleton, 2000) and the theoretical work by Milankovitch (1920). First monitoring efforts focused on recording the changing positions of glacier termini, starting in the 17th century and systematically organized in the late 19th century, for example by the German and Austrian Alpine Club (Fritzsch, 1898; Groß, 2018). In the case of catastrophic glacier advances, as reported several times during the Little Ice Age, for instance, for Vernagtferner in the Ötztal Alps (Nicolussi, 2012), local observers often reported the velocity of terminus advances over short periods. At that stage of development, glaciological theory and monitoring techniques, the monitoring of horizontal ice flow velocities was already well established for Alpine glaciers. Stone line velocities were recorded at 5 glaciers among 20 glaciers regularly monitored for length changes in the Eastern Alps, for example, at the glaciers of Pasterze (Nicolussi and

Patzelt, 2001), Vernagtferner (Braun et al. 2012) or Hintereisferner (Span et al., 1997), or in the Western Alps at Rhonegletscher (Mercanton, 1916; Roethlisberger, 1963) or Mer de Glace (Berthier and Vincent, 2012). At the glaciers Unteraargletscher and Mer de Glace, ice flow was already measured during the 1840s (Forbes, 1846; Agassiz, 1847).

For Alpine glaciers, monitoring of velocity records received less attention after the turbulent decades of the First and Second World War. Development of glaciological programmes focused on hydrological programmes and mass balance programmes, as the understanding of glacier flow was advanced by deformation measurements on Hintereisferner and in the theoretical work of Finsterwalder (1907) and Hess (1924).

Estimates of the global glaciers' contribution to sea level rise is one of the urgent topics of research (e.g. Jacob et al., 2012; Zemp et al., 2019), and estimates of the state of regional glacier inventories are needed. Glacier flow velocities which can be derived from remote-sensing data are an important parameter that provides essential information on dynamic response, which is part of the mass balance evolution of a glacier. For example, ELA (Equilibrium Line Altitude), which played a major role in large-scale data collections on global climate change, has been observed to be above summits and thus undefined for Eastern Alpine glaciers for much of the last decade (WGMS, 2017).

In this paper, two long-term series and two series of about a decade of ice flow velocities are revisited and compared with classical in situ mass balance measurements (Hoinkes, 1970) and ALS (Airborne Laser Scanning) data (Abermann et al., 2010). On these four glaciers, ice flow velocities are measured in situ at stone lines (horizontal velocities) or at stakes (3D velocities). The stones and stakes are annually relocated to their original position. Long-term velocity data are recorded annually, shorter time series also reveal seasonal variabilities. The four glaciers are also part of a network of long-term measurements of glacier fluctuations (Groß, 2018); area and volume change have been recorded in inventories from the LIA (Little Ice Age) maximum onwards (Patzelt, 1980; Groß, 1987; Kuhn et al., 2012; Fischer et al., 2015).

This paper presents ice flow velocity records on well investigated mountain glaciers and their relation to other in situ monitoring parameters. This data can be used for validation of numerical ice flow models and further research on ice flow velocity as monitoring parameter and climate proxy on various scales.

## 2 Glacier sites and data

The ice flow velocities have been recorded on four of the largest glaciers in the Ötztal Alps (Austria). Hintereisferner (HEF), Kesselwandferner (KWF), Gepatschferner (GPF) and Taschachferner (TSF) are neighbouring glaciers (Fig. 1) differing in size, aspect and elevation ranges (Tab. 1).

HEF, a typical valley glacier, has a long tradition of hydrological, meteorological, geophysical and glaciological investigations (e.g. Blümcke und Hess, 1899; Förtsch and Vidal, 1956; Hoinkes and Steinacker, 1975; Kuhn et al., 1999; Fischer, 2010; Helfricht et al., 2014; Strasser et al., 2018). Ice flow velocities on HEF have been sporadically measured at ablation stakes but almost annually for more than 100 years at stone lines (Span et al., 1997). Line 3 on HEF is the oldest stone line, started in 1895 and lasted until 1985 when the glacier retreated upstream from the location of the profile. Records

at Line 6 were started in 1932/33 and at Line 7 in 2013/14. In situ mass balances have been measured since 1952 (Hoinkes, 1970; Fischer, 2010; Fischer et al. 2013; Strasser et al., 2018).

The investigations on KWF are historically linked to those on HEF (e.g. Kuhn et al., 1985) with the same long-term investigations of length variations since 1884 and mass balances since 1952 (Fischer et al., 2014). The terminus of KWF detached from the tongue of HEF in 1914. KWF is a plateau glacier and covers a small elevation range compared to HEF, GPF and TSF, especially since the terminus retreated to the top of a steep terrain level and detached from the dead ice body in the front of this step in 2012. Velocity measurements were started in 1965 by Schneider (1970) at ablation stakes and at accumulation stakes along the centre flow line of the glacier. A comparison between direct glaciological and geodetic measurements on KWF as well as on HEF was presented by Fischer (2011). The main outcome of the comparison was that both mass balance data sets generally agree within measurement uncertainties.

GPF is the second largest glacier of the Austrian Alps. The main glacier rests on a wide but hilly plateau and the tongue descends through a narrow valley. After early first mappings (Sonklar, 1860; Finsterwalder, 1928), GPF was subject to geophysical investigations (Giese, 1963), photogrammetric analyses (Keutterling and Thomas, 2006) and is one of the Eastern Alpine key research sites, with extensive knowledge on its Holocene fluctuations (Nicolussi and Patzelt, 2001). Recently, Gepatschferner became part of a detailed study on geomorphodynamics (Heckmann et al., 2012; Heckmann and Morche, 2019). In this study, the stake network at the glacier tongue was extended from three stakes, where velocities have been measured since 2009, to 16 stakes in 2012. TSF is similar to GPF a plateau and valley glacier with a wide accumulation area and a narrow glacier tongue. The stake velocity records on TSF were started together with those on GPF in 2009 at three positions at the glacier tongue.

On both glaciers, GPF and TSF, the positions of the stakes are measured several times during summer months, allowing a discussion of the sub-seasonal variability. In contrast, the velocity records at HEF and KWF were performed once a year and can be discussed in relation to their long-term mass balance records.

## 3 Methods

Based on the historical development of geodetic techniques, different methods came into operation on these glaciers during the past century. Trigonometric networks were installed in 1894 on HEF (Blümcke and Hess, 1899) and in 1965/66 on KWF (Schneider, 1970) to determine glacier surface velocities with a theodolite at stone lines on HEF and ablation stakes on both glaciers (Fig. 1). On HEF and KWF, stake velocities were measured using a theodolite and tachymeter until 2009. Since then DGPS (Differential Global Positioning System) have been used. On GPF and TSF, the full series was measured by DGPS (System: Topcon, Antennas: Hiper V, Software: Magnet Tools).

The velocity records are compared to direct glaciological and geodetic mass balance measurements from Hoinkes (1970), Schneider (1970), Fischer (2010), Fischer et al. (2013), Stocker-Waldhuber et al. (2017) and Strasser et al. (2018) on HEF, KWF and GPF. In these publications, the surface mass balances were derived from stakes and snow pits by using the direct

glaciological method (Hoinkes, 1970). Additionally, DEMs (Digital Elevation Models) and DODs (DEMs of Difference) from photogrammetric or high-resolution ALS data came into operation to determine volume and elevation changes (Abermann et al. 2010).

### 3.1 Velocity measurements at stone lines

The method of stone lines (Heim, 1885; Hess, 1904) was used only on HEF at three cross-profiles. The position of several stones and their distance to each other is fixed within a defined cross-profile. The number of stones depends on the glacier width and thus varies in time with any expansion or reduction of the glacier. The position of the stones was measured initially with tachymetric systems and since 2009 with DGPS. The stones are flat with a diameter ranging from 0.15 m to a maximum of 0.3 m. The distance between the original defined position of the stone within the profile and the position in the

subsequent year is measured using a measuring tape. The horizontal displacement is calculated in consideration of the elevation change or the slope of the surface at each stone.  The stones are then moved to their original position. From 2009, the displacement was calculated from the measured DGPS positions, but the measuring tape is still used for control. The annual velocities at the stone lines are given as the mean annual values of the stones in the profile and thus depends on the number of stones.

Velocity records from ablation stakes complemented earlier data for Line 6 (before 1932/33) and Line 7 (before 2013/14) for periods when the stakes were reinstalled at their original position. The stakes are located at the central flow line of the glacier, thus representing the maximum flow velocity at the profile. A ratio of 80% between the mean velocity from the stone line and the maximum velocity at stakes located at the centre of profile (Span and Kuhn, 2003) was taken to compare the stake values with the mean values from the stone lines.

### 3.2 Velocity measurements at stakes

Velocity measurements are performed at ablation stakes and at accumulation stakes on KWF, GPF, TSF and complement the stone lines on HEF. The position of the stakes and their motion on KWF is measured at the top of the stake and calculated to the lower end of the stake, its base point.. This has the advantage that the measured velocity is not affected by surface changes of accumulation or ablation. Figure 2 by Schneider (1970) shows the components of the velocity vector (d) at the

base point of the stake within the accumulation area (left side) and the ablation area (right side) between two points in time (t1, t2) depending on submergence (negative value of v) and emergence (positive value of v). This definition coincides with the definition of submergence and emergence in Cogley et al., (2011). The vertical motion can be calculated as the remainder of the absolute elevation change of the surface ($\Delta d$) and the accumulation or ablation ($\Delta a$) or from the elevation change due to the sloping surface ($\Delta h$) and the vertical component ($\Delta z$) of the velocity vector (d) (Schneider, 1970). The difference

between the actual flow path (d), which is the length of the velocity vector, to the horizontal motion ($\Delta s$), which is the projected velocity, results from the vertical component ($\Delta z$). Annual values of the horizontal flow velocity ($\Delta s/a$) as well as the vertical motion values of submergence and emergence were calculated for 365 days and on the basis of a fixed

coordinate system. The horizontal velocity component (s) and the vertical component (v) in Schneider (1970) correspond to the definitions of (u) and (w) in Cuffey and Paterson (2010). Upward motion is positive.

The stakes on KWF are reinstalled annually at their initial (xy) position. Redrillings and measurements are conducted with a level rod for exact perpendicular conditions. The reflector or the DGPS antenna is directly mounted to the top of the stake.

Therefore duraluminium stakes with rigid connection are used on KWF as ablation stakes (Ø = 2 cm) and thicker accumulation stakes with a diameter of 5 cm for the necessary resistance against snow pressure. Tipping over and melting in may lead to errors in measuring the vertical motion. On KWF, a tipping over of the stakes is avoided by the use of duraluminium stakes. Wood wool underneath the stake and at the downhill side of the stake protects against melting in.

## 4 Accuracies and uncertainties

The investigations on KWF are accurate to the lower cm-level (0.05 to 0.1 m) for the horizontal and the vertical stake position determined with theodolite and tachymeter. This is possible because of statistical adjustment of the measured positions using resection and intersection techniques and especially the statistical adjustment of the trigonometric network. Further details on the derived errors for this specific dataset can be found in Schneider (1970). In 2009, DGPS measurements with RTK (real time kinematic) procedure came into operation. During these measurements, the base-station is located in

close proximity to the glacier at fix-points of the trigonometric network (Schneider, 1970; Niederwald, 2009; Weide, 2010; Zauner, 2010), allowing a staking-out of the stake position with a comparable accuracy in the lower cm-level. The overall accuracies depend on a proper implementation of the measurement methods. The precision requirements on KWF are in the range of ±0.05 m per single measurement or at least ±0.1 m for the period between the two readings for the horizontal and the vertical displacement.

In contrast to the stakes on KWF, ice flow velocities at GPF and TSF are measured at wooden ablation stakes with DGPS and post-processing procedure, the measured positions refer to the glacier surface. The minimum occupation time for DGPS with post-processing is 10 minutes but is mostly in the range of 30 to 60 minutes. The nearest base station, which provides basic data within the GPS and GLONASS satellite system, is located in Malles (STPOS-MABZ, 46°41'9.55" N, 10°33'3.73" E, South Tyrol, Italy). The baseline to the stakes on GPF is about 26 km and about 32 km to the stakes on TSF.

The accuracies from the post-processing procedure on these glaciers are ±0.1 m per single measurement or ±0.2 m for the period between two readings at its best. Additionally, the distance to the base station and shading effects of the surrounding topography lead to higher uncertainties for the DGPS measurements in the lower decimetre to metre-level.

The uncertainty of the stone line measurements on HEF, determined with a measuring tape, can only be estimated. They depend on the measuring distance, surface roughness and possible slipping of stones on the ice surface. The number of

stones in the profile varies in time, which leads to a systematic error to the mean velocity of the lines. The number of stones at Line 7 decreased from initially 19 stones to 17 stones and at Line 6 to 4 stones in 2017. The velocities of the stone lines refer to the horizontal component of the velocity vector (Δs) and are calculated from theodolite and tachymeter

measurements until 2009 and with RTK-DGPS since then with an absolute error of ±0.05 m per single measurement. An estimated uncertainty of the mean stone line velocity of 5% of the annual displacement is higher in contrast to the measurements on KWF due to the possible slipping-motion of stones in the profile. The slipping error mainly depends on the surface slope and the annual surface ablation and depends less on the surface velocity. This means, the higher the ablation rate and the steeper the surface slope, the higher the slipping error. Consequently, this error becomes more important at low velocities (Blümcke and Finsterwalder 1905).

## 5 Results

### 5.1 Hintereisferner

Three stone line records on HEF display the variation of glacier surface velocities for different periods, in total for more than 100 years (Fig. 3). Three periods with increasing surface velocities were recorded on HEF. The first and most extensive acceleration of surface velocity happened before 1920, with a maximum mean stone line velocity of 125 m per year at line 3 in 1919 (Hess, 1924). This is 112 m per year above the mean flow velocity of 13 m per year of the long-term average (1895-1985) at this location. The velocity increased to this maximum within a few years and decreased very quickly until 1922 resulting in a small advance of the glacier terminus in subsequent years of around 60 m (Span et al. 1997). The second period was recorded from 1935 to the early 1940s and the most recent one during the 1970s. During that time, the mass balance of the glacier was positive for several years (Fig. 3). Since 1980, surface velocities on HEF have continuously decreased at the stone lines to about 4 m per year in the most recent years at Line 6, and to about 7-8 m per year at Line 7. This continuous decrease is accompanied by strong negative mass balances in the most recent decades.

### 5.2 Kesselwandferner

On KWF measurements were started in the hydrological year 1965/66, including horizontal and vertical ice flow velocities (Fig. 4 and Fig. 5). These long-term investigations document different glacier states at a longitudinal profile of up to ten accumulation and ablation stakes. There are two main contrasting periods, the first one from the start of the measurements to 1985, and the second period since then. During the first period, the glacier advanced because of positive mass balances. During that time 75% of the measured glaciers in Austria and Switzerland advanced due to positive mass balances as a result of decreasing summer temperature and increasing annual precipitation (Patzelt, 1985). The surface velocity of the glacier increased, but with decreasing magnitudes from the terminus (L10) to the uppermost stake (L1) within the accumulation area. This means an increased velocity gradient along the glacier, with maximum of about 90 m per year at the terminus declining to a few metres per year at the highest elevations. The gradient of the vertical velocities was also large, with a submergence of up to 3 m per year within the accumulation zone to an emergence of up to 5 m per year at the lowermost stake. During that time the ELA (equilibrium line altitude) shifted to lower elevations which can be seen as the transition from submergence to emergence from stake to stake.

The advancing state of the glacier ended in 1985, followed by a sharp decrease of the surface velocities and a reduction of the velocity gradients along the flow line. The terminus velocity response was large due to the positive mass balances. A significant change in velocity near the ELA at L6 did not occur until negative mass balances occurred around 1985. Submergence transitioned gradually to emergence around 1990 at stakes L8 and L7 and in 2005 at L6, representing a shift of the ELA to higher elevations. During that time, ELA shifted from 3130 m (mean value from 1985 to 1990) to 3214 m during the period 2005-2010 and even above the crest level in 2003 for the first time since the beginning of the mass balance measurements (Fischer et al. 2014; Strasser et al. 2018). At the lowermost stake L10, velocity decreased rapidly to almost nil because of the decreasing mass supply to the terminus. This area became ice-free in 2010 (Fig. 4 and Fig. 5). The latest positive mass balance occurred in 2015, with an immediate response in the horizontal and vertical velocities.

## 5.3 Gepatschferner and Taschachferner

The measurements of ice flow velocities and ablation on GPF and TSF were started in 2009/10. The stake network on GPF was extended in 2012. During these measurements, interannual velocity fluctuations were small, especially at the three stakes on TSF (54, 55 and 56). At the lowermost stake 54, the horizontal velocities were less than 10 m per year during the whole period. The two higher-altitude points 55 and 56 returned velocities of 30 to around 40 m per year. The higher values at 55 compared to 56 are caused by topographic effects, with a steepening of the glacier tongue and a narrowing of the cross section from stake 56 towards stake 55 (Fig. 6).

On GPF a general trend of decreasing surface velocities was found at all stakes, with a larger decline in velocity at the terminus than at the upper cross-profile (71-75). At stake 62, a funnel-shaped surface depression, caused by an evacuation of subglacial sediments due to a heavy precipitation event, led to a spatially limited increase of surface velocity in that area and a later decrease to almost nil (Stocker-Waldhuber et al., 2017). In total, a general slowdown of velocities at the tongue of GPF was found since the beginning of the measurements.

An example of subseasonal fluctuations in surface velocity is given in Figure 7 for stake 65 at GPF, which has been measured from 2009 with least data gaps since then. The velocities are given as mean values per day to make the different time periods of the stake readings comparable. During the winter seasons velocities generally decrease. Maximum values were typically found in August each year, except for the years 2013 and 2014 with earlier peaks in July. During winter months the elevation change of the surface from geodetic measurements is close to zero or even positive, while surface velocity is decreasing. The opposite process is found during the summer season, when the highest surface velocities go along with the most negative elevation change.

## 6 Discussion

The investigations show that the magnitude of the fluctuations of the surface velocities is higher at the ablation stakes compared to those within the accumulation area and highest at the lowermost stakes. This means that changes in observed

velocity, especially at ablation stakes, are very well suited for documenting the glacier state, even more so at a fast reacting glacier like KWF. This is supported by a linear regression of annual mean specific balance (b) of the total glacier area of HEF and KWF versus the mean specific balance of their accumulation areas (bc) for the period 1965/66 – 1999/00 by Span and Kuhn (2003). They found nearly identical correlation coefficients for the two balances, while the standard deviation σ(b) was twice as high as σ(bc), documenting the higher sensitivity of the ablation areas to mass balance changes of the two glaciers.

On GPF and TSF these measurements were conducted exclusively at stakes at the tongue where the decreasing velocity rates represent the retreating state of the glacier. The decreasing velocities were found especially on GPF, in contrast to TSF, where the flow rates remained almost constant during these investigations. This is caused by the specific topographic conditions of the TSF glacier tongue, which is steeper and shorter, and the terminus is located at a higher altitude compared to the tongue of GPF.

The subseasonal fluctuations on GPF (Fig. 7) represent the typical acceleration of the glacier during summer months, which is well known and was, for example, already measured on HEF between the end of July and mid-September from 1900 to 1904 by Blümcke and Finsterwalder (1905). The summer peaks on GPF and TSF were found mainly during August each year. These subseasonal fluctuations depend on the drainage network which is driven by the amount of melt water, seasonal and extreme weather conditions (Iken, 1977; Gudmundsson, 2002). The accuracy of the measurements on GPF varies over time and depends on shading effects of the surrounding topography. For example, the peak in 2013 shows the greatest uncertainty due to topographic shading effects and is thus not representative for the actual surface velocity.

Generally, the glacier surface velocities will be overestimated by stone velocity measurements especially at low velocities and high ablation rates. This difference is shown by Hess (1924) for the period between 1913/14 and 1921/22 with measured surface velocities at neighbouring stones and boreholes.

## 7 Conclusion

The long-term investigations of the surface velocities at these glaciers document the state of each glacier and its response to a climate signal. Three periods with accelerating velocities caused by positive mass balances were found in the longest time series on HEF. A time shift of the maximum values from higher to lower stone line profiles indicates the response time of the tongue. Despite the increase in surface velocities during these three periods, mass gain on HEF was insufficient for the terminus to advance, except for a small advance during the 1920s. The 1920s peak velocity at line 3 was also confirmed by measurements at boreholes, stakes and additional stones within the ablation area (Hess, 1924). In contrast to HEF, the terminus of KWF advanced by more than 250 m from the 1970s to the 1980s (Patzelt, 1985; Fischer et al., 2018). KWF presents an immediate response at all profiles concurrently, which means that a mass gain or increase of the thickness within the accumulation area causes an increase of the emergence at the lowermost stakes within one year.

During glacier retreat the transition from submergence to emergence shifts to higher elevations, as was found on KWF from 1986 to the present. As a consequence, the magnitude of the vertical velocities decreases, which leads to increasing retreat rates of the terminus but at the same time to an increase of the thickness at higher elevations in case of a positive mass balance. Apart from the effect of the mass balance, according to the continuity equation, local thickness changes can also be caused by convergent or divergent glacier flow.

For the investigated temperate mountain glaciers, ice flow velocity is a glaciological parameter that reacts very quickly to changes in the forcing as, for example, the glacier mass balance variability (Huss, 2012). As conventional parameters like ELA tend to be above summit for the investigated glaciers under current conditions and specific mass balance is affected by rapid changes in area, long-term monitoring of ice flow provides valuable additional information on the glacier state. Our data set allows to develop and validate perspectives on ice flow velocity as monitoring tool.

**Data availability**

Velocity data of the four glaciers Hintereisferner, Kesselwandferner, Gepatscherner and Taschachferner are available at https://doi.pangaea.de/10.1594/PANGAEA.896741 (Stocker-Waldhuber et al., 2018). New data will be added every year.

**Competing interests**

The authors declare that they have no conflict of interest.

**Acknowledgements**

Maintaining long-term monitoring is always a challenging task and requires financial support and the help of numerous people, to whom we would like to gratefully express our thanks. Terminus variations of the glaciers and the velocity records on HEF relate to the annual measurements of the Austrian Alpine Club. Mass balance terms are provided by the World Glacier Monitoring Service (WGMS) and the Institute of Atmospheric and Cryospheric Sciences (ACINN). Research (project PROSA) on GPF was funded by DFG (SCHM 472/16-1, SCHM 472/17-2 and BE 1118/33-3) and FWF (I 894-N24 and I 1646-N19) and funding continues from glacier ski resort Kaunertaler Gletscher GmbH and Tiroler Wasserkraft AG (TIWAG), which also supports the measurements on TSF. We want to thank H. Schneider who started the velocity records on KWF and continued them for more than 50 years. These records are now supported by the non-profit organisation Glacier and Climate. We want to thank B. Scott for editing the English. We thank Reinhard Drews (Ed.) and the reviewers Mauri Pelto and Andreas Bauder, who gave us useful comments, which helped us to improve this paper.

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

**Table 1: Geographic location and characteristic numbers of GPF, HEF, KWF and TSF from Austrian Glacier Inventory 3 of 2006 (Fischer et al. 2015) and the year of the start of velocity measurements (meas. since). exp: exposition, Sc: accumulation area, Sa: ablation area.**

| name | location | exp. Sc | exp. Sa | altitude range [m] | area [km²] | meas. since |
|------|----------|---------|---------|--------------------|-----------|-------------|
| GPF | 46.85°N, 10.75°E | NE | N | 2116-3501 | 16.62 | 2009/10 |
| HEF | 46.79'N, 10.75°E | E | NE | 2436-3715 | 7.49 | 1894/95 |
| KWF | 46.84°N, 10.79°E | SE | E | 2754-3496 | 3.82 | 1965/66 |
| TSF | 46.90°N, 10.86°E | N | NW | 2424-3756 | 5.71 | 2009/10 |

**Figure Captions**

Figure 1: Location of the stone lines (3, 6 and 7) on Hintereisferner (HEF) and stakes on Kesselwandferner (KWF), Taschachferner (TSF) and Gepatschferner (GPF). On Gepatschferner, stakes 60 to 66 are longitudinal stakes from the glacier snout upwards to the first cross profile including the stakes 67 to 70 (from the orographic right to the left). Stakes 71 to 75 are located in a cross profile at the root zone of the tongue. The glacier area was taken from the Austrian Glacier Inventories (GI) from LIA (little ice age) around 1850, GI1 from 1969, GI2 from 1998 and GI3 from 2006 (Fischer et al.,

2015). Background: Orthophoto from 2015; data source: Land Tirol – data.tirol.gv.at.

Figure 2: Drawings by Schneider (1970) of the motion of a stake and changes at the glacier surface (Oberfl.) between two time steps (t1, t2) within the accumulation area (left) and the ablation area (right). d: flow path (length of the velocity vector), v: vertical velocity, Δs: horizontal velocity (projected velocity), Δd: absolute surface elevation change, Δa: point mass balance (relative surface elevation change from accumulation or ablation).

Figure 3: The mean annual velocities of the stones at lines 3, 6 and 7 on Hintereisferner since 1894/95 (= 1895). Data series extended since Span et al. (1997) and annual specific surface mass balance (b direct) since 1953 (Strasser et al., 2018; WGMS, 2017; original data: Hess, 1924) as well as the geodetic balances from DoDs (b geodetic) by Fischer (2011). Location of the stone lines s. Fig. 1.

Figure 4: Annual horizontal flow velocities (Δs/a) at the accumulation and ablation stakes on Kesselwandferner (e.g. the year

2015 refers to the hydrological year 2014/2015) and the specific surface mass balance (b direct) (Strasser et al., 2018; WGMS, 2017) as well as the geodetic balances from DoDs (b geodetic) by Fischer (2011). Location of the stakes s. Fig. 1.

Figure 5: Annual vertical velocities (Δv/a) at the accumulation and ablation stakes on Kesselwandferner (e.g. the year 2015 refers to the hydrological year 2014/2015). Positive values represent emergence flow, negative ones represent

submergence flow. Location of the stakes s. Fig. 1.

Figure 6: Annual horizontal flow velocities (Δs/a) on Gepatschferner and Taschachferner and L9 at Kesselwandferner for comparison (e.g. the year 2015 refers to the hydrological year 2014/2015). GPF (a): Selection of the longitudinal stakes

at the tongue of Gepatschferner. GPF (b): Three stakes at the cross profile; location s. Fig. 1: 71: orogr. left, 73: centre, 75: orogr. right.

Figure 7: Mean daily horizontal velocities (Δs/day) at stake 65 on Gepatschferner between the measurements as an example of the subseasonal fluctuation of surface velocity. The peak in July 2013 shows the highest uncertainty, very likely because of few satellites due to shading effects of the surrounding topography, which depends on the time of the measurements. Additional information is given by the mean elevation change per day from ALS DoDs at the position of the stake (data extended from Stocker-Waldhuber et al. (2017)).

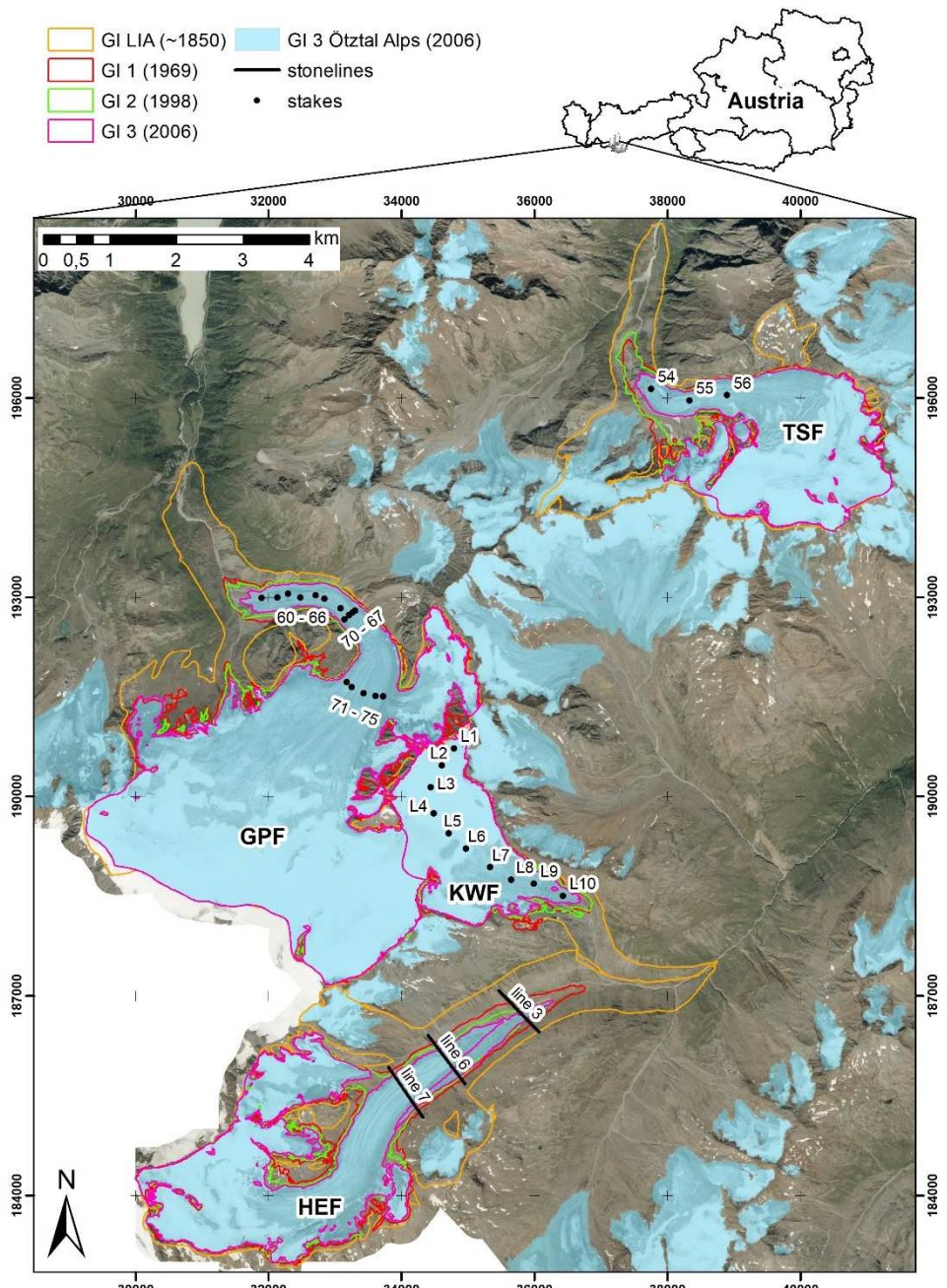

**Figure 1: Location of the stone lines (3, 6 and 7) on Hintereisferner (HEF) and stakes on Kesselwandferner (KWF), Taschachferner (TSF) and Gepatschferner (GPF). On Gepatschferner, stakes 60 to 66 are longitudinal stakes from the glacier snout upwards to the first cross profile including the stakes 67 to 70 (from the orographic right to the left). Stakes 71 to 75 are located in a cross profile at the root zone of the tongue. The glacier area was taken from the Austrian Glacier Inventories (GI) from LIA (little ice age) around 1850, GI1 from 1969, GI2 from 1998 and GI3 from 2006 (Fischer et al., 2015). Background: Orthophoto from 2015; data source: Land Tirol – data.tirol.gv.at.**

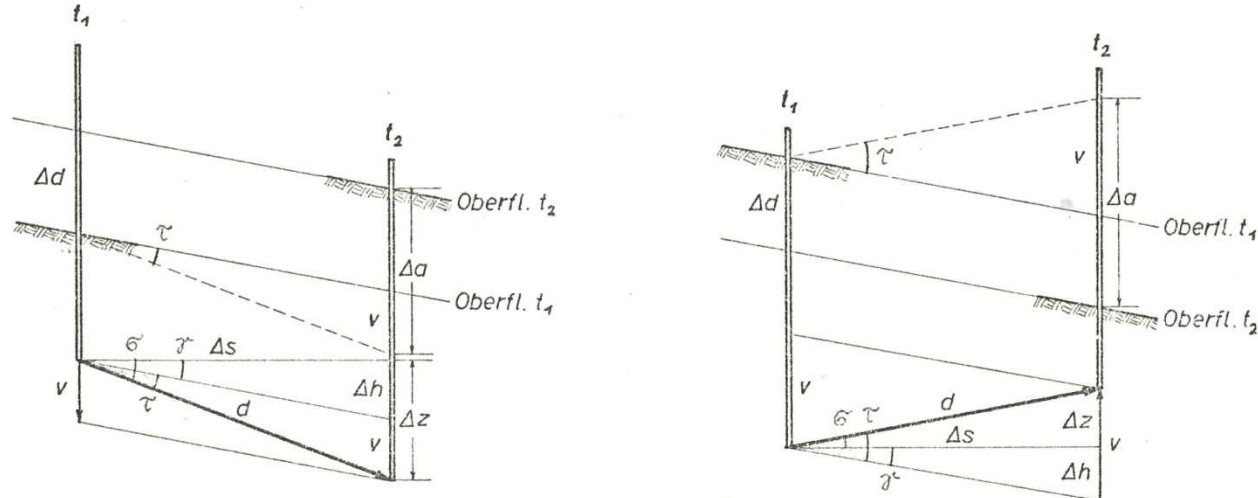

**Figure 2: Drawings by Schneider (1970) of the motion of a stake and changes at the glacier surface (Oberfl.) between two time steps (t1, t2) within the accumulation area (left) and the ablation area (right). d: flow path (length of the velocity vector), v: vertical velocity, Δs: horizontal velocity (projected velocity), Δd: absolute surface elevation change, Δa: point mass balance (relative surface elevation change from accumulation or ablation).**

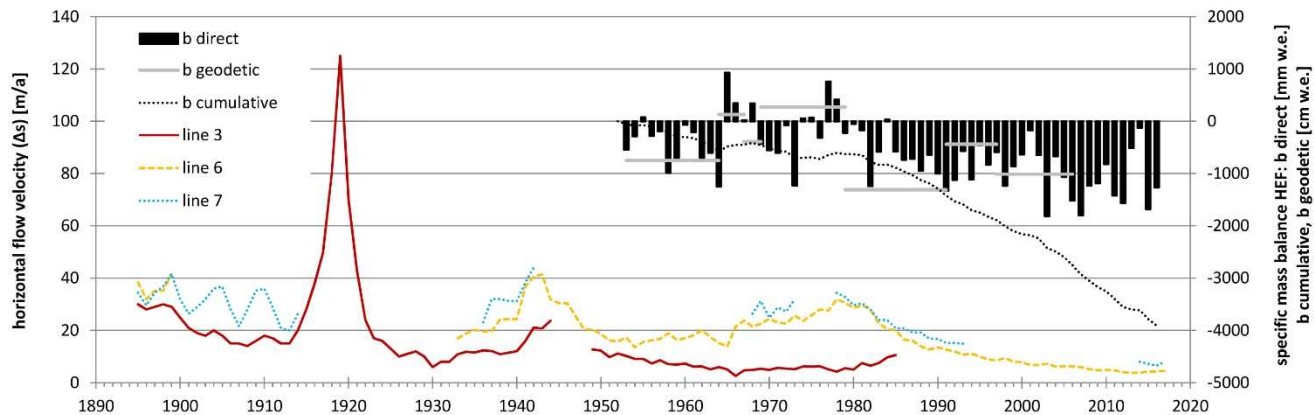

**Figure 3: The mean annual velocities of the stones at lines 3, 6 and 7 on Hintereisferner since 1894/95 (= 1895). Data series extended since Span et al. (1997) and annual specific surface mass balance (b direct) since 1953 (Strasser et al., 2018; WGMS, 2017; original data: Hess, 1924) as well as the geodetic balances from DoDs (b geodetic) by Fischer (2011). Location of the stone lines s. Fig. 1.**

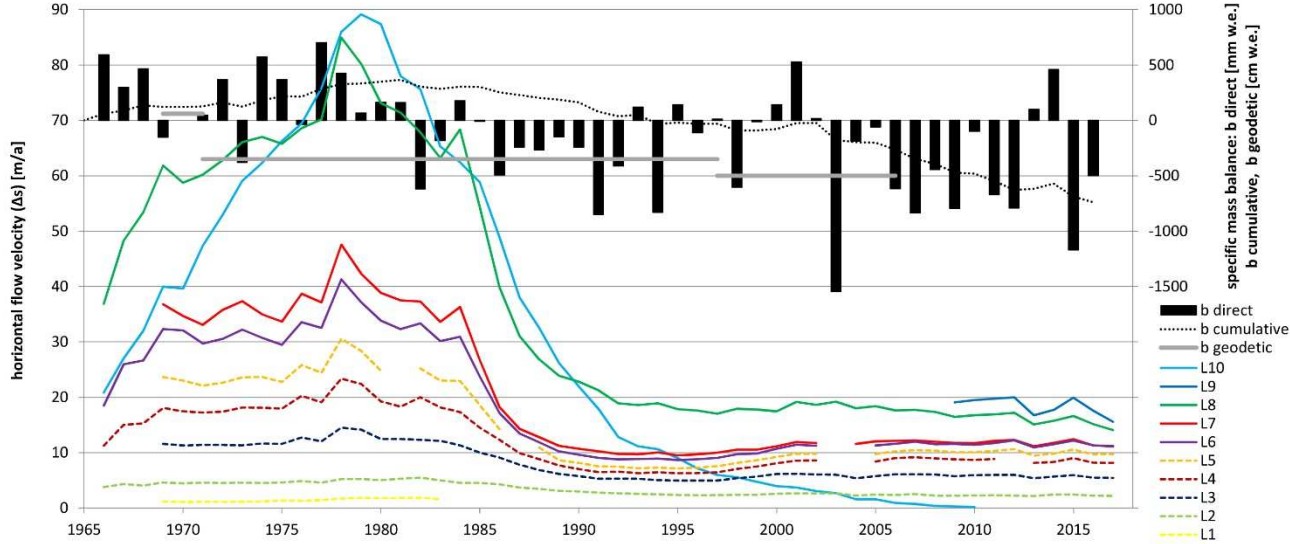

**Figure 4: Annual horizontal flow velocities (Δs/a) at the accumulation and ablation stakes on Kesselwandferner (e.g. the year 2015 refers to the hydrological year 2014/2015) and the specific surface mass balance (b direct) (Strasser et al., 2018; WGMS, 2017) as well as the geodetic balances from DoDs (b geodetic) by Fischer (2011). Location of the stakes s. Fig. 1.**

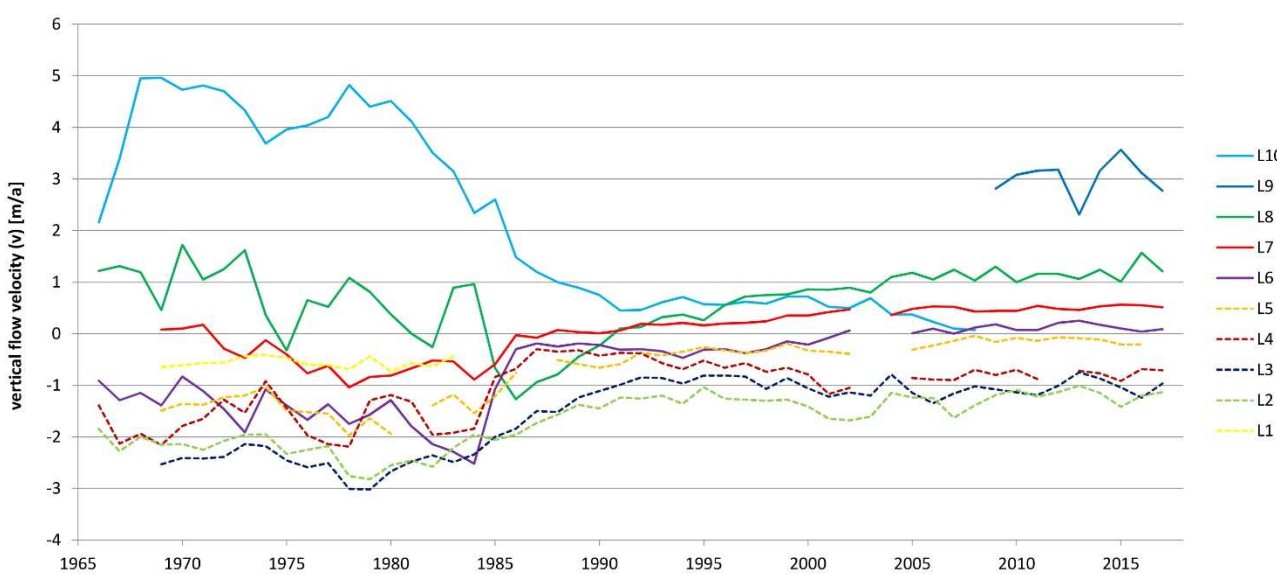

**Figure 5: Annual vertical velocities (Δv/a) at the accumulation and ablation stakes on Kesselwandferner (e.g. the year 2015 refers to the hydrological year 2014/2015). Positive values represent emergence flow, negative ones represent submergence flow. Location of the stakes s. Fig. 1.**

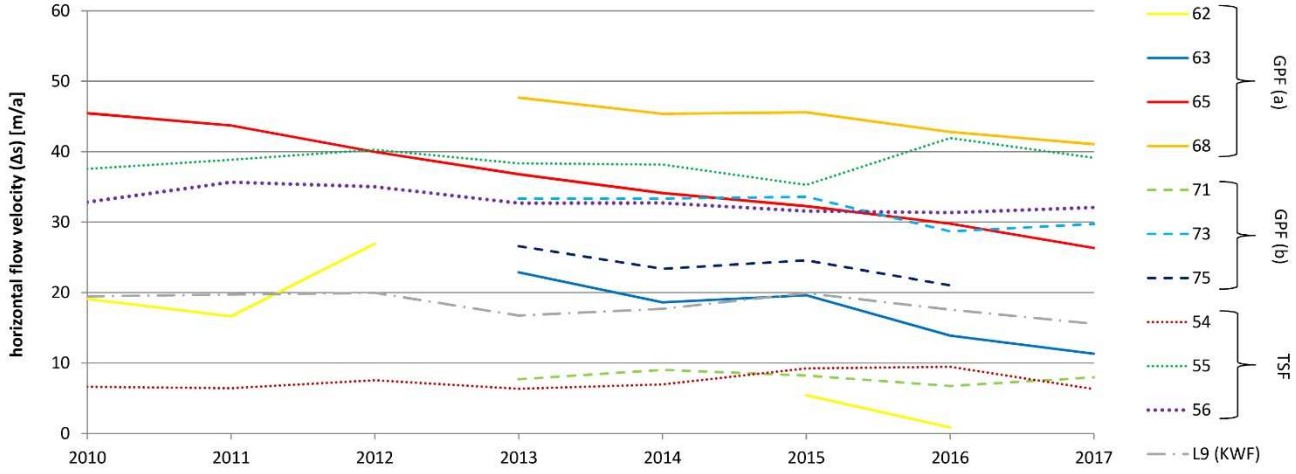

**Figure 6: Annual horizontal flow velocities (Δs/a) on Gepatschferner and Taschachferner and L9 at Kesselwandferner for comparison (e.g. the year 2015 refers to the hydrological year 2014/2015). GPF (a): Selection of the longitudinal stakes at the tongue of Gepatschferner. GPF (b): Three stakes at the cross profile; location s. Fig. 1: 71: orogr. left, 73: centre, 75: orogr. right.**

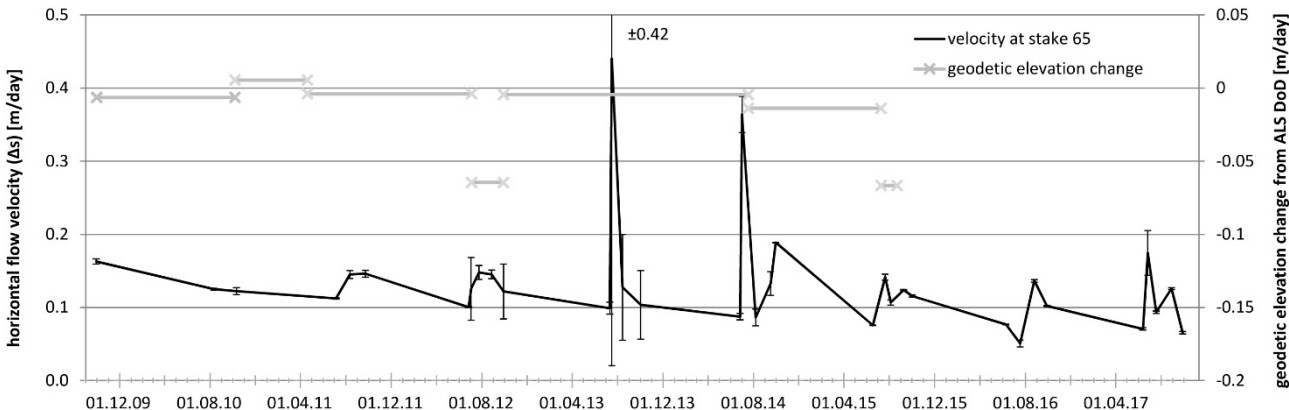

**Figure 7: Mean daily horizontal velocities (Δs/day) at stake 65 on Gepatschferner between the measurements as an example of the subseasonal fluctuation of surface velocity. The peak in July 2013 shows the highest uncertainty, very likely because of few satellites due to shading effects of the surrounding topography, which depends on the time of the measurements. Additional information is given by the mean elevation change per day from ALS DoDs at the position of the stake (data extended from Stocker-Waldhuber et al. (2017)).**

