# Peer review of "Long-term records of glacier surface velocities in the Ötztal Alps (Austria)"

_Earth System Science Data, 2018_

## Referee Comment (RC1) · Pelto (Referee) · 2 Feb 2019

Stocker-Waldhuber et al (2019) provide the context for a rare long term glacier velocity record in the Ötztal Alps. Because mass balance records exist on some of the glaciers for significant periods it is evident that velocity change is useful for identifying responses to climate caused mass balance change. This type of record is an important data set to report. The specific comments below are focused primarily on clarity. I encourage more scrutiny of the stone line velocity error assessment on HEF. Also the 1919 velocity on HEF is that plausible?

2-6: "...and estimates of the state of regional glacier inventories are needed, glacier flow velocities which can be derived from remote sensing data are an important parameter that provides essential information on dynamic response, which is part of the mass balance evolution of a glacier."

2-21: Replace "but" with "that"

2-31: provide a descriptive sentence on KWF similar to that for GPF, the main glacier rests on a wide but hilly plateau and the tongue descends through a narrow valley.

3-10: provide a descriptive sentence on TSF similar to that for GPF.

3-30: What is the size range of the stones?

4-11: . . . calculated to the lower end of the stake, its base point."

4-24: define better what is "lower cm-level" 5-10 cm?

5-17: The error assessment on HEF should be better stated. That 5% is used is okay, what is this in terms of cm per year for the most rapid areas of motion? I do wonder if the 5% is realistic for stone slippage, or too large at areas of rapid motion, slippage should not be that different based on velocity alone for example stone slippage in 1980 at 5 m per year is much different than for the 1920's maximum of 120 m, yet the stone slippage mechanism should have changed little.

5-24: Is this maximum velocity for a point or a line?

5-25: The increase in velocity implies a major mass balance change, based on other observed and reported changes in this record, and would suggest more than a minor advance would occur. Here or in the discussion could you identify why the terminus change or mass balance changes was not as significant as the velocity change would imply. What was the velocity in 1918 and 1920? If the change to 1919 is really large is that acceleration plausible or the ensuing deceleration?

6-8: The two following sentences conflate actual velocity and temporal velocity change. Be consistent in reporting the difference in the second sentence. "The surface velocity of the glacier increased, but with decreasing magnitudes from the terminus (L10) to

the uppermost stake (L1) within the accumulation area. This means an increased velocity gradient along the glacier, with maximum of about 90 m per year at the terminus declining to a few metres per year at the highest elevations."

6-14: Worth commenting on the velocity response in Figure 4 where terminus velocity response was large simply to declining positive mass balance after 1978, and a significant change in velocity near the ELA at L6 did not occur until negative mass balances occurred around 1985.

6-16: Reference for the higher ELA, should provide a quantity for this shift as well.

6-21: There is an insignificant velocity decline through time on TSF (54-56) which is contrast to the other glaciers, why?

6-26: "... with a larger decline in velocity at the upper profile (71-75) than at the terminus."

7-22: To identify state is it not the deviations in velocity that identify changes in state, simply the actual velocity measurements at a moment in time would not be useful in determining the state of an unknown glacier " This means that changes in observed velocity, especially at ablation stakes........"

7-32: The conclusion of peak in velocity in the summer deserves closer definition and referencing. Most alpine glaciers have a velocity peak sometime early-mid summer as the drainage network matures, and a decline late in summer. The extent to which any of these four differ from this is important to note. Given the annual data for HEF and KWF is may only be GPF and TSF where such comment can be made.

8-5: The rapid response in terms of velocity is documented in studies that look at terminus response time of glaciers which lag both velocity and mass balance. There is useful response time data for the Alps that can be cited here ie. Huss (2012). This would enhance the value of the statement and the methods applied here.

8-5: The following statement is incomplete and not accurate, please modify. In fact

ELA is sometime above the summits, surface mass balance observations still provide an accurate measure. The date that the transient snow line goes above the glacier is also a measure that provides value. "As conventional parameters like ELA tend to be above summit for the investigated glaciers under current conditions and specific mass balance is affected by rapid changes in area."

Figure 4-6: Each of these figures have numerous time series that simply are hard to distinguish with gray scale lines. A color scheme is recommended which can be based on zone of the glacier as well.

Huss, M.: Extrapolating glacier mass balance to the mountain-range scale: the European Alps 1900–2100, The Cryosphere, 6, 713– 727, doi:10.5194/tc-6-713-2012, 2012

---

## Referee Comment (RC2) · Andreas Bauder (Referee) · 7 Mar 2019

The paper presents a data-set of surface flow velocities measurements on 4 glaciers in Austria ranging from more than 100 years of observations on Hintereisferner to a decade on Gepatsch- and Taschachferner. Velocity fluctuations are interpreted in terms of glacier wide mass balance and length fluctuations.

General comments:

Indeed, ice flow is an important property of glacier and this parameter has got surprisingly low attention in monitoring programs. Ice flow velocity depends on ice thickness and surface slope. So ice thickness change is most suitable for interpretation of velocity variations. I do understand that surface elevation was measured as long with the posi-

tion of the flow markers, and thickness change can be determined (as an example see Fig 5.3 of the latest Glaciological Report http://doi.org/10.18752/glrep_137-138). Sure the surface topography is a result of mass balance but with some dynamical response and local ice thickness is more appropriate than glacier wide balance quantities. Moreover, I would recommend - if shown - to plot the cumulative mass change rather than annual values.

The method sections suffers from two shortcomings. (1) A systematic bias results when calculating a mean of a variable number of measurements. I see two potential alternatives - central or maximum value as well as average of a constant, fixed subset of measurements. (2) Although the difference of emergence/submergence and the vertical component of the velocity vector are introduced in detail, throughout the paper (e.g. Fig.5) a misleading terminology of vertical velocity for the emergence/submergence motion. Vertical velocity is only valid with regard to a fixed coordinate system. Emergence/submergence is the motion relative to the surface resulting as a an apparent vertical displacement.

The effect of melting in and tipping over of flow markers is not addressed. Important with regard for the accuracy/uncertainty is the fact that vertical movement is one order of magnitude lower than the horizontal component and moreover of the same order as the counteracting processes of mass balance and thickness change. So any uncertainty of any of these may affect all.

Your interpretation and discussion makes extensive use of length variation. It would be more convincing for the reader if you would plot this information - at least for some glaciers (e.g. HEF)

General quality of the Figures is relatively poor and therefore hard to read. Probably this is just a minor problem of Figures generated in vectorized format that have been transformed with a poorly resolved raster format when inserted to the manuscript? Labels are all fuzzy, rather small and gradients between different lines difficult to separate.

Checking of the online data-set prepared for download on pangea.de was not possible, because access was denied. I made several unsuccessful attempts.

Detailed/minor points (indicated by page.line):

1.12: Ice age theory was earlier established by Agassiz (Alps) or Lyell (UK) already back in the first half of 19th century. Penck & Brueckner may have confirmed the theory later on.

1.18: I miss proper references of first systematic ice flow measurements in the Alps in the 1840s on Unteraargletscher and Mer de Glace (Agassiz, 1847; Forbes, 1846).

1.21: I recommend to use the official spelling of 'Rhonegletscher' to be consistent with all the other mentioned glaciers

1.21: Berthiere -> Berthier

2.10: I do not agree this paper presents 2 long-term series and 2 series of only about a decade.

2.11: Acronym ALS was not yet introduced

3.14: Acronym DGPS not introduced

5.2: rod level -> level rod

7.3: Unclear what is the 'expected inverse process'? Surface elevation change may result from both processes melting or a dynamic adjustment.

7.11-12: This statement is not correct - velocity variation is a direct response to thickness change as a result of the climatic forcing where as the terminus fluctuations is delayed and damped by the dynamic adjustment. Both are sensitive!

7.31-32: The reason for summer speed-up has been well investigated e.g. Iken, 1978 or Gudmundsson, 2002

Fig.1: Replacing the GI 1, GI 2 and GI 3 labels with the respective years would be more

reader friendly.

15.5: pint mass balance is the right terminology for delta_a and more appropriate.

Fig.3: are you sure that the individual, extremely high value of 1919 is correct? Are there any arguments against an outlier?

Fig.7: awkward ticks / tick interval -> using quarters would be easy to read

References:

Agassiz (1847), Système glaciaire ou recherches sur les glaciers, leur mécanisme, leur ancienne extension et le rôle qu'ils ont joué dans l'histoire de la terre. Première partie: Nouvelles études et expériences sur les glaciers actuels, leur structure, leur progression et leur action physique sur le sol, V. Masson, Paris.

Forbes (1846), Illustrations of the viscous theory of glacier motion, Philosophical Transactions of the Royal Society, 136(1), 143-210.

Gudmundsson, G. (2002), Observations of a reversal in vertical and horizontal strain-rate regime during a motion event on Unteraargletscher, Bernese Alps, Switzerland. Journal of Glaciology, 48(163), 566-574. doi:10.3189/172756502781831043

Iken (1978), Variations of surface velocities of some Alpine glaciers measured at intervals of a few hours. Comparison with Arctic Glaciers, Zeitschrift Gletscherkunde, 13(1/2): 23-35.

---

## Author Comment (AC1) · 20 Mar 2019

The comment was uploaded in the form of a supplement:
https://www.earth-syst-sci-data-discuss.net/essd-2018-153/essd-2018-153-AC1-
supplement.pdf
* * *

---

## Author Response (AR1)

The review is cited underlined; answers of authors are formatted with indents:

> We thank Mauri Pelto and Andreas Bauder for their important comments and suggestions, which helped us to improve this manuscript. We fully agree with the comments and we considered them point by point.
>
> Regarding the 1919 velocity on HEF: The velocity increased to the maximum within a few years and decreased very quickly until 1922 (m/a 1915-1922: 28, 38, 50, 80, 125max (1919), 70, 43, 24) resulting in an advance of the glacier tongue by about 60 m (Span et al. 1997). We will add this information in more detail.
>
> We consider the direct velocity measurements plausible. They were performed, despite of WWI, by the same team of observers twice a year. Although measurements in the firn area could not be done during war, the measurements in the ablation area have been continued. The measurements include deep drilling holes as well as stone lines (see Tab. 6 from Hess, 1924). The observers (Hess, 1924) did use the same network of fixed points as before and after the acceleration. They critically reflected the velocity increase and found additional points showing that the velocity increase is plausible.
>
> In 1914, Kesselwandfener lost connection to Hintereisferner, but advanced again afterwards. The terminus position was mapped in 1917, 1918, 1919 and 1920. The velocity increase, which was also found remarkably by Hess, triggered a cartographic survey of the glacier surface with 6 stereo pairs, using the fixed point network. Hess (1924) found indications for increasing glacier mass potentially triggering the velocity increases in 1919. The lowering of the surface elevation at the terminus coincides with the decreasing velocities.
>
> With respect to the state of science at the time of measurements, the time series can be considered amongst the most reliable and best documented world wide. We will add additional data and information from Hess (1924) to the text.

Tab. 6    Geschwindigkeiten in Meter/Jahr.

| | Höhe m | 1913/14 | 1914/15 | 1915/16 | 1916/17 | 1917/18 | 1918/19 | 1919/20 | 1920/21 | 1921/22 |
|---|---|---|---|---|---|---|---|---|---|---|
| 40-m-Loch | 2408 | 11,5 | 23,0 | 18,3 | 25,7 | 45,8 | — | — | — | — |
| Untere ziegelrote Linie, Punkt 7 | 2438 | 20,0 | 29,0 | 25,7 | 29,9 | 51,3 | — | — | — | — |
| Grüne Linie, Punkt 8 | 2527 | 28,9 | 44,2 | 41,4 | 55,5 | 77,8 | 109,1 | 68,8 | 49,4 | 13,5 |
| Untere rote Linie, Punkt 9 | 2564 | 33,9 | | 90,2 | 64,1 | 84,3 | 126,6 | 72,7 | 44,1 | 21,8 |
| 153-m-Loch | 2557 | 30,8 | | 85,6 | 63,4 | 83,8 | 124,8 | 60,1 | 36,5 | 23,2 |
| 214-m-Loch | 2570 | 29,4 | | 90,4 | 65,1 | 82,2 | 121,5 | 64,0 | 44,0 | 22,3 |
| Mittlere blaue Linie, Punkt 8 | 2600 | 33,1 | | 103,8 | 63,4 | — | — | — | 48,2 | — |
| Obere ziegelrote Linie, Punkt 10 | 2635 | 35,0 | ←—189,5—× | | —296,2——→ | | | — | — |
| Blaugrüne Linie, Punkt 9 | 2689 | 37,8 | | 109,0 | 80,8 | 94,7 | 123,0 | 91,1 | 46,5 | 14,2 |
| Bohrloch von 1910 | 2685 | — | — | — | 75,1 | 97,5 | 112,0 | 70,2 | 46,1 | 13,8 |
| Obere blaue Linie, Punkt 9 | 2720 | ←——240,3—× | | —302,2——→ | | | | — | — |
| Stange β | 2747 | 37,1 | ←—187,8—→ | | | 81,5 | 118,6 | 77,9 | — | — |
| Dreikant II | 2811 | 44,6 | ←————————379,3————————→ | | | | | | 57,5 | — |

Hess, H.: Der Hintereisferner 1893 bis 1922. Ein Beitrag zur Lösung des Problems der Gletscherbewegung. Zeitschrift für Gletscherkunde, 13, 145-2013, 1924.

Span, N., Kuhn, M., and Schneider, H.: 100 years of ice dynamics of Hintereisferner, Central Alps, Austria, 1884-1994. Annals of Glaciology, 24, 297-302. doi:10.1017/S0260305500012349, 1997.

Referee #1: Mauri Pelto

Stocker-Waldhuber et al (2019) provide the context for a rare long term glacier velocity record in the Ötztal Alps. Because mass balance records exist on some of the glaciers for significant periods it is evident that velocity change is useful for identifying responses to climate caused mass balance change. This type of record is an impor-tant data set to report. The specific comments below are focused primarily on clarity. I encourage more scrutiny of the stone line velocity error assessment on HEF. Also the 1919 velocity on HEF is that plausible?

2-6: "…and estimates of the state of regional glacier inventories are needed, glacier flow velocities which can be derived from remote sensing data are an important parameter that provides essential information on dynamic response, which is part of the mass balance evolution of a glacier."

Done, as suggested

2-21: Replace "but" with "that"

Done, as suggested

2-31: provide a descriptive sentence on KWF similar to that for GPF, the main glacier rests on a wide but hilly plateau and the tongue descends through a narrow valley.

Done, as suggested

3-10: provide a descriptive sentence on TSF similar to that for GPF.

We added the descriptive information on KWF and TSF.

3-30: What is the size range of the stones?

The stones are flat with a diameter ranging from 0.15 m to a maximum of 0.3 m. We added this information.

4-11:…calculated to the lower end of the stake, its base point."

Done, as suggested

4-24: define better what is "lower cm-level" 5-10 cm?

We added "(0.05 to 0.1 m)"

5-17: The error assessment on HEF should be better stated. That 5% is used is okay, what is this in terms of cm per year for the most rapid areas of motion? I do wonder if the 5% is realistic for stone slippage, or too large at areas of rapid motion, slippage should not be that different based on velocity alone for example stone slippage in 1980 at 5 m per year is much different than for the 1920's maximum of 120 m, yet the stone slippage mechanism should have changed little.

The slipping motion mainly depends on the ablation rate and the surface slope. We added this information.

5-24: Is this maximum velocity for a point or a line?

This is given for a line, added.

5-25: The increase in velocity implies a major mass balance change, based on other observed and reported changes in this record, and would suggest more than a minor advance would occur. Here or in the discussion could you identify why the terminus change or mass balance changes was not as significant as the velocity change would imply. What was the velocity in 1918 and 1920? If the change to 1919 is really large is that acceleration plausible or the ensuing deceleration?

For point measurements and the mean value for line 3 (Fig. 3). (cf. comments above and table 6 by Hess (1924)). We added this information in the results and discussion sections

6-8: The two following sentences conflate actual velocity and temporal velocity change. Be consistent in reporting the difference in the second sentence. "The surface velocity of the glacier increased, but with decreasing magnitudes from the terminus (L10) to the uppermost stake (L1) within the accumulation area. This means an increased velocity gradient along the glacier, with maximum of about 90 m per year at the terminus declining to a few metres per year at the highest elevations."

Done, as suggested

6-14: Worth commenting on the velocity response in Figure 4 where terminus velocity response was large simply to declining positive mass balance after 1978, and a significant change in velocity near the ELA at L6 did not occur until negative mass balances occurred around 1985.

We added these comments on the different responses.

6-16: Reference for the higher ELA, should provide a quantity for this shift as well.

We gave values on the rising ELA on KWF (Fischer et al. 2014; Strasser et al. 2018)

Fischer, A., Markl, G., Schneider, H., Abermann, J., and Kuhn, M.: Glacier mass balances and elevation zones of Kesselwandferner, Ötztal Alps, Austria, 1952/1953 to 2012/2013. PANGAEA, doi:10.1594/PANGAEA.818757, 2014.

Strasser, U., Marke, T., Braun, L., Escher-Vetter, H., Juen, I., Kuhn, M., Maussion, F., Mayer, C., Nicholson, L., Niedertscheider, K., Sailer, R., Stötter, J., Weber, M., and Kaser, G.: The

Rofental: a high Alpine research basin (1890–3770 m a.s.l.) in the Ötztal Alps (Austria) with over 150 years of hydrometeorological and glaciological observations, Earth Syst. Sci. Data, 10, 151-171, https://doi.org/10.5194/essd-10-151-2018, 2018.

6-21: There is an insignificant velocity decline through time on TSF (54-56) which is contrast to the other glaciers, why?

> This is related to the specific topographic conditions of TSF. We added this information to the discussion.

6-26: "…with a larger decline in velocity at the upper profile (71-75) than at the terminus."

> We wrote "…with a larger decline in velocity at the terminus than at the upper cross-profile (71-75)."

7-22: To identify state is it not the deviations in velocity that identify changes in state, simply the actual velocity measurements at a moment in time would not be useful in determining the state of an unknown glacier "This means that changes in observed velocity, especially at ablation stakes……."

> We changed to "… changes in observed velocity…"

7-32: The conclusion of peak in velocity in the summer deserves closer definition and referencing. Most alpine glaciers have a velocity peak sometime early-mid summer as the drainage network matures, and a decline late in summer. The extent to which any of these four differ from this is important to note. Given the annual data for HEF and KWF is may only be GPF and TSF where such comment can be made.

> We added details on the summer peak at GPF and TSF and added the following references: Iken, 1977 or Gudmundsson, 2002 (c.f. comment below by A. Bauder)

8-5: The rapid response in terms of velocity is documented in studies that look at terminus response time of glaciers which lag both velocity and mass balance. There is useful response time data for the Alps that can be cited here ie. Huss (2012). This would enhance the value of the statement and the methods applied here.

> Thank you, we added this citation.

8-5: The following statement is incomplete and not accurate, please modify. In fact ELA is sometime above the summits, surface mass balance observations still provide an accurate measure. The date that the transient snow line goes above the glacier is also a measure that provides value. "As conventional parameters like ELA tend to be above summit for the investigated glaciers under current conditions and specific mass balance is affected by rapid changes in area."

> We deleted the rest of the sentence ("…monitoring of ice flow can be recommended as additional surveyed parameter at mountain glaciers").

Figure 4-6: Each of these figures have numerous time series that simply are hard to distinguish with gray scale lines. A color scheme is recommended which can be based on zone of the glacier as well.

We changed to colour schemes

Huss, M.: Extrapolating glacier mass balance to the mountain-range scale: the European Alps 1900–2100, The Cryosphere, 6, 713– 727, doi:10.5194/tc-6-713-2012,2012

Referee #2: Andreas Bauder

The paper presents a data-set of surface flow velocities measurements on 4 glaciers in Austria ranging from more than 100 years of observations on Hintereisferner to a decade on Gepatsch- and Taschachferner. Velocity fluctuations are interpreted in terms of glacier wide mass balance and length fluctuations.

General comments:

Indeed, ice flow is an important property of glacier and this parameter has got surprisingly low attention in monitoring programs. Ice flow velocity depends on ice thickness and surface slope. So ice thickness change is most suitable for interpretation of velocity variations. I do understand that surface elevation was measured as long with the position of the flow markers, and thickness change can be determined (as an example see Fig 5.3 of the latest Glaciological Report http://doi.org/10.18752/glrep_137-138). Sure the surface topography is a result of mass balance but with some dynamical response and local ice thickness is more appropriate than glacier wide balance quantities. Moreover, I would recommend - if shown - to plot the cumulative mass change rather than annual values.

> We added the cumulative specific mass balance to Figure 3 (HEF) and Figure 4 (KWF).

The method sections suffers from two shortcomings. (1) A systematic bias results when calculating a mean of a variable number of measurements. I see two potential alternatives - central or maximum value as well as average of a constant, fixed subset of measurements. (2) Although the difference of emergence/submergence and the vertical component of the velocity vector are introduced in detail, throughout the paper (e.g. Fig.5) a misleading terminology of vertical velocity for the emergence/submergence motion. Vertical velocity is only valid with regard to a fixed coordinate system. Emergence/submergence is the motion relative to the surface resulting as a an apparent vertical displacement.

> ad 1: We added comments on the number of stones and gave the current number of stones in the profiles. Naturally, when a glacier tends to become narrower, with the same distance between stones the number of stones reduces. This is not necessarily a shortcoming. Keeping the number of stones constant by reducing the distance between the stones would not end up necessarily in a higher accuracy, as the variability between neighbouring stones is low, as the stones are not positioned in a crevasse zone.

> ad 2: We rephrased the figure 5 caption. We added the information on the fixed coordinate system in the methods section.

The effect of melting in and tipping over of flow markers is not addressed. Important with regard for the accuracy/uncertainty is the fact that vertical movement is one order of magnitude lower than the horizontal component and moreover of the same order as the counteracting processes of mass balance and thickness change. So any uncertainty of any of these may affect all. Your interpretation and

discussion makes extensive use of length variation. It would be more convincing for the reader if you would plot this information - at least for some glaciers (e.g. HEF) General quality of the Figures is relatively poor and therefore hard to read. Probably this is just a minor problem of Figures generated in vectorized format that have been transformed with a poorly resolved raster format when inserted to the manuscript? Labels are all fuzzy, rather small and gradients between different lines difficult to separate.

Duraluminium stakes are used against tipping over, wood wool underneath and at the downhill side is used against melting in. We added the information to the accuracy section. Tipping over may occur at wooden ablation stakes, therefore vertical motion is not calculated at these stakes.

We improved the Figures resolution and script sizes.

Checking of the online data-set prepared for download on pangea.de was not possible, because access was denied. I made several unsuccessful attempts.

The dataset is now published open access.

Detailed/minor points (indicated by page.line):

1.12: Ice age theory was earlier established by Agassiz (Alps) or Lyell (UK) already back in the first half of 19th century. Penck & Brueckner may have confirmed the theory later on.

It is an old discussion in science theory if a hypothesis is a valid theory before being confirmed by empirical evidence. We agree that there has been a bunch of discussions with respect to fluvial and glacial transport, and left more space for important earlier contributions to the idea of glacial transport. We added the Reference on Agassiz (1847) and that the theory was confirmed by Penck and Brückner (1909).

1.18: I miss proper references of first systematic ice flow measurements in the Alps in the 1840s on Unteraargletscher and Mer de Glace (Agassiz, 1847; Forbes, 1846).

Thank you, we added these references.

1.21: I recommend to use the official spelling of 'Rhonegletscher' to be consistent with all the other mentioned glaciers

Done, as suggested

1.21: Berthiere -> Berthier

Done

2.10: I do not agree this paper presents 2 long-term series and 2 series of only about a decade.

We changed to "two long-term series and two series of about a decade"

2.11: Acronym ALS was not yet introduced

Done

3.14: Acronym DGPS not introduced

Done

5.2: rod level -> level rod

Done

7.3: Unclear what is the 'expected inverse process'? Surface elevation change may result from both processes melting or a dynamic adjustment.

We deleted the sentence "A comparison with the geodetic elevation change…" and wrote in the next sentence: "During winter months the elevation change of the surface form geodetic measurements is close to zero…"

7.11-12: This statement is not correct - velocity variation is a direct response to thickness change as a result of the climatic forcing where as the terminus fluctuations is delayed and damped by the dynamic adjustment. Both are sensitive!

Thank you. It is true that both are sensitive. We deleted this sentence.

7.31-32: The reason for summer speed-up has been well investigated e.g. Iken, 1978 or Gudmundsson, 2002

We added this information and the two references, Iken 1977 and Gudmundsson 2002.

Fig.1: Replacing the GI 1, GI 2 and GI 3 labels with the respective years would be more reader friendly.

We added the respective years to the Figure legend.

15.5: pint mass balance is the right terminology for delta_a and more appropriate.

We added "point mass balance"

Fig.3: are you sure that the individual, extremely high value of 1919 is correct? Are there any arguments against an outlier?

This Peak was also observed at boreholes, stakes and stones (Hess, 1924); we added this information. See the extensive material above.

Fig.7: awkward ticks / tick interval -> using quarters would be easy to read

We changed the tick interval to quarters

References:

[revised manuscript text omitted]

---

## Author Response (AR2)

Answers of the authors are formatted with indents:

**Topical Editor Decision: Publish subject to minor revisions (review by editor)** (17 Apr 2019) by

Reinhard Drews
Comments to the Author:
Dear authors,

thank you for responding to the reviews in detail. The reviewers have raised some important points, and the revised manuscript has certainly improved. There is no doubt that you have compiled an important dataset, that deserves publication as it is outstanding particularly regarding the temporal coverage. However, I am not yet convinced by structure and presentation of the paper. Below you will find a long list of editorial and structural changes that I suggest. I am not a native english speaker myself, and I don't require that all editorial remarks are put into practice.

However, particularly section 4 and section 6 require revision in terms of ordering, clarity, and succinctness. Details of the GNSS measurements are not yet clear to me, and also the derivation of errors (and their temporal variability) is not rigorously enough described for a paper in ESSD.

Note that none of my comments require further data analysis, and I echo the reviewer's judgement that the dataset is scientifically meaningful. Another round of revisions with a focus on clear structure and presentation style will give this dataset the framework it deserves.

Kind regards, Reinhard Drews

> Thank for your comments and especially your detailed suggestions which helped us to improve the manuscript. Our editorial office did another language check, and we added the suggested information.

General Comment: Please provide a version including track-changes next time.

> Please find the track-change version at the end of this response

P1, l3: Suggestion of rephrasing: "Here we present a unique dataset of yearly-averaged ice-flow velocity measurements at stakes and stone lines covering more than 100 years on Hintereisferner and more than 50 years on Kesselwandferner. Moreover, the dataset contains subseasonal variations of ice flow from Gepatschferner and Taschachferner covering almost 10 years."

> Done, as suggested

P1, l 13 - Awkward opening sentence with "established", "confirmed", "later confirmed" and "explained". Suggestion: "…after Agassiz (1847) hypothesized the theory of ice ages, which was then confirmed by Penck and Brückner (1909) and further substantiated with isotope analysis on polar ice cores (REF?) and theoretical work by Milankowitch (REF?)"

> Done, as suggested and we added the three references:

Hays, J.D., Imbrie, J., and Shackleton, N.J.: Variations in the Earth's Orbit: Pacemaker of the Ice Ages, Science, 194 (4270) 1121-1132, 1976.

Milankovitch, M.: Théorie Mathématique des Phénomènes Produits par la Radiation Solaire (Gauthier-Villars, Paris), 1920.

Shackleton, N.J.: The 100,000-Year Ice-Age Cycle Identified and Found to Lag Temperature, Carbon Dioxide, and Oribtal Eccentricity, Science, 289 (5486), 1897-1902, 2000.

P1, l15: The monitoring efforts in the 17th and 19th century require a reference, or at least some details on who and where.

We added the information on the measurements of the Austrian Alpine Club

Fritzsch, M.: Verzeichnis der bis Sommer 1896 in den Ostalpen gesetzten Gletschermarken. Verlag des Deutschen und Österreichischen Alpenvereins, Wien, 1898

P1, l 20: Split than sentence up into two. It is unnecessarily convoluted.

Done

P2, l 8: Replace Jacob et al., 2012 with more recent Zemp et al, 2019?

We added Zemp et al. 2019

Zemp, M., Huss, M., Thibert, E., Eckert, N., McNabb, R., Huber, J., Barandun, M., Machguth, H., Nussbaumer, S.U., Gärtner Roer, I., Thomson, L., Paul, F., Maussion, F., Kutuzov, S., and Cogley, J.G. (2019): Global glacier mass changes and their contributions to sea-level rise from 1961 to 2016. Nature, 568, 382-386, doi:10.1038/s41586-019-1071-0

P2, l 8: Again a compound-complex sentence structure containing two add-ons with "which" and one inset with ", and estimates … ,". This needs to be disentangled because it is unnecessarily hard to read.

Splitted into two sentences

P2, l 10: "For example ELA.." -> "For example, ELA…"

Done

P2, l 14: I think "ALS" is more commonly referred to as "Airborne Laser Scanning" (not Scan)

Done

P2, l16: "at annual resolution" -> "annually"

Done

P2, l16: Make two sentences. Second one "The stones advect with ice flow ( i.e. recording ice-flow in a Lagrangian system). The stakes, on the other hand, are relocated every year to their original position (i.e. recording ice-flow in an Eulerian system).

We added: The stones and stakes are annually relocated to ….

P2, l17 "net" -> "network" and then later "from THE Little Ice Age onwards" .

Done

P2, l20 Remove "Without further discussion…." and start with "This paper presents ice-flow velocity records on well …."

Done

Close the introduction with one sentence on how this data can be applied (maybe as a baseline dataset for using glacier length/velocity as proxy for climate change over centennial timescales ?)

Added: This data can be used for validation of numerical ice flow models and further research on ice flow velocity as monitoring parameter and climate proxy on various scales.

P2, l 24: "Ötztal Alps (Austria)" later, "that differ" -> "differing"

Done

P2, l 30 "behind" > "upstream"

Done

P3, l 8: What was the main outcome of this comparison?

Added: The main outcome of the comparison was that both mass balance data sets generally agree within measurement uncertainties.

P3, l 19: several measurements during summer do not allow discussion of seasonality (but sub-seasonal variations during summer or variability between different summers).

We changed to "sub-seasonal"

P3, l26 ". After than DGPS was used."

Done

P4, l10 Ok, forget my previous comment about Lagrange and Euler, but I hope you see how this was written confusingly.

cf. comment above

P4, l11 : What is the approximate lateral variation in a stone line, and why do you "only" report the average value? Also "depends on the number of stones" is correct but does not give credit to the reviewer comment (referring to a statistical bias) nor to the lateral variability (simply speaking, if all stones are located at the same place the mean does not depend on thenumber of stones). Rephrase and add details.

The approximate lateral variation in a stone line depends on the variation of the glacier width. The stones are positioned equidistant within the cross profile. If the glacier shrinks, thus the number of stones decrease. As evident from the Figure below, the glacier width at line 6 at Hintereisferner shrunk by about 50 % between 1953 and 2006.

[Figure]

The stones are positioned with a distance of 25 m each along a straight line (cross-profile). So we can expect to cover the range of velocity variability by sampling at these equidistant locations.

Regarding the statistical bias: Mathematically speaking, we can do a simple model presuming a sinus function of velocity:

[Figure]

The mean of the dense sampling (2m) data is 0.6295, the mean of the less dense sampling (20m) data is 0.6301, so the difference is 0.09% only.

In this example, the glacier shrinks from 180 to 100 m.

[Figure]

Sampling still is equistant (20 m. orange), but the number of measurement points reduced as distance between sampling points is constant and total length reduced. Now the constant sampling rate of 20 m gives a mean of 0.512, i.e. 22% error. The velocity curve is smoother than the sinus, the velocity at the glacier boundaries are >0, and maximum velocity reduces when the profile length decreases counteracting the effect of reducing samples in terms of relative error.

| length | distance | number points | mean | difference | percent |
|---|---|---|---|---|---|
| 180 | 2 | 91 | 0.62956002 | | |
| 180 | 20 | 9 | 0.63014242 | -0.00058241 | -0.09251008 |
| 100 | 2 | 51 | 0.62393169 | | |
| 100 | 20 | 6 | 0.51294726 | 0.11098443 | 21.6366162 |

P4 l15: It is unclear to me where this ratio comes from, why it was used for comparison and what the take away message is.

The 80% refers to the maximum velocity at the centre flow line of the glacier. This means 80% of the stake velocity is comparable to the mean velocity of the profile as found by Span and Kuhn 2003.

P4. L 24: Whether velocities are positive or negative depends on your choice of coordinate system (some people to z vertically downwards). Be precise here and don't outsource such conventions to an external reference. Also add " relative to the surface" for emergence/submergence velocities.

We calculate with a fix coordinate system, velocities from the lowest point of the stake upwards are positive. This is given in the text: P4 L27 and P5 L3

P4, l 30 "correlate" -> "correspond"

Done

P5 , l 1 "Aim at high accuracies" ? Rephrase: Measurements done with theodolites and tachymeters are accurate to the lower cm-level. This is possible, because of statistical adjustment …. . Also add: Further details on the derived errors for this specific dataset can be found in Schneider, 1979.

We rephrased this part

P5 , l 1 I admit that I don't know what "resection" means in this context and my online translation tool only finds medical terms. Please rephrase or explain for the wider ESSD audience.

Resection and intersection means „Rückwärts- und Vorwärtseinschneiden". We added „… of the measured positions using resection and intersection techniques…"

P5 l 7 what is "staking-out" ?

Staking-out means the locating of a given position (Abstecken)

P5 l 1: First: Collecting all uncertainties in one section is a good idea. However, then this section should only contain this and details of the individual methods (e.g. what stakes are used, the use of wood wool, what rigid connections were used, where the base station was located and so on) should be transferred to the sections 3.1- 3.2. Second: This section needs more structure. What I expect here (given the section heading) would be something like:

Measurements based on theodolits and tachymeters are with XX cm. This depends on a,b,c and details can be found here. Differences between the observed glaciers are a,b,c :

Measurements based on DPGs are within XX cm. This depends on a,b,c. Differences between the observed glaciers are a,b,c.

So for KWF you used your own base station and for GPF & TSF a base station in Italy. Why?

For KWF, the fix points for the reference station are in use since 1962. There is no such network for GPF and TSF, and the station in Italy is near (and was installed much later than 1962, so that it was no option for KWF).

P5, l19 State-of-the art GNSS base stations should receive not only GPS and GLONASS, but also Galileo, Beidou and so on. Is there a reason why you specifically mention GPS and GLONASS ?

Yes, that's true, but so far only the GPS and GLONASS systems were used within the time series presented here. With the ongoing monitoring Galileo will be included.

P5 l20 "linear distance" – "baseline"

Done

P5, l25 what is the order of magnitude for observe sub-seasonal velocities in summer, and how do these relate to the errors of up to 1 m?

The errors of up to 1m are caused by shading effects, which depend on daytime due to the changing satellite coverage. On short timescales a great error from one measurement can be greater than the velocity, or the calculated distance between two readings. Sentence deleted

P5, 25 The use of RTK – DGPS is unclear to me. RTK to me means "real-time kinematic" and requires a radio-link between base station and rover (I doubt this was the case when using the base station in

Malles). Also, why would you use RTK for measuring the stake positions? Wouldn't it be better to determine the position in the post-processing ? Why is the stake position required in real-time?

> RTK-DGPS is used on KWF and needed because of the exact relocation (staking-out) of the stakes at their initial positions as this position is fixed. The post-processing procedure is used on GPF and TSF.

Following up on this: I am not convinced by the details given regarding the DGPS data. Additionally to the confusion whether the position was determined during post-processing or in real-time, it is also unclear to me what processing scheme has been applied. For example, how did you get the position of your own local base station? In my experience those details are not unimportant when aiming for an absolute horizontal precision of less than 5 cm. Moreover, there is no distinction between errors in horizontal and in vertical direction. Shouldn't it be more difficult to get the vertical displacement? If all of these points result in smaller errors compared to other sources (e.g. the tilt of the stakes, or slipping of stones) this is ok but should be explicitly stated and quantified.

> We added the information on the post-processing procedure and added references on the fix points of the local base stations.
>
> Niederwald, T.: Festpunktbestimmung mit GPS für Gletscher-Monitoring-Projekte in den Ötztaler Alpen (Tirol), Master thesis, Hochschule für Angewandte Wissenschaften, FH München, Fakultät für Geoinformation, 2009.
>
> Weide, S.: Estimating the Height Accuracy of Airborne Laser Scanning with GPS and Calculation of Glacier Movement at Hintereisferner, Master thesis, Hochschule für Angewandte Wissenschaften FH München, Fakultät für Geoinformation, 2009
>
> Zauner, R.: Glaziologische Analyse der Gletscheroberfläche am Blockgletscher Äußeres Hochebenkar, Hintereisferner und am Kesselwandferner (Ötztaler Alpen), Bachelor thesis, Hochschule für Angewandte Wissenschaften FH München, Fakultät für Geoinformation, 2010.

p.5 l 30: GPS measurements should be characterized with an absolute error. 5 % is additional useful information but requires the local flow velocity for context.

> Added: with an absolute error of ±0.05 m per single measurement

Section 4 is important, but it is currently too unstructured and partially also unclear. This needs to be revised.

P6, l 10 Suggestion add: This is XX m/a (or XX %) above the mean flow velocity measured at this location.

> Done

P6 l 20 "During the first period, " (comma) and then later "advanced" (instead of "was in an advancing state")

> Done

P6 l 21 "presented positive length changes" sounds complicated. Why not "advanced"?

We changed to: "… advanced due to positive…"

P6 l 32 would it be better to say "Submergence transitioned to emergence around 1990…"

Done, as suggested

P8 l 11 "and highest ant the lowermost stakes"

Done

Section 6 has a quite narrative character and I am not a friend of merging Discussions and Conclusions. Many people only read abstract and conclusion of a paper (and they tend not read the conclusions if they are intertwined with discussion elements). Consider taking these two sections apart, and provide a succinct conclusion emphasizing the uniqueness of your dataset and possible applications.

We splitted section 6 into two sections

P8 l 26: That the accuracy of measurements depends on topographic shading has already been mentioned above. Also it is not quantified. I wonder if this statement is important enough to be repeated in the conclusion.

It is unclear to me, why the "peak in 2014" for GPF has the highest uncertainty. What event in time makes this measurement less certain than the others? Section 4 estimates errors at GPF to be within pm 20 cm "at its best". I don't see how this explains the variability in errors shown in Figure 7.

This peak was caused by topographic shading effects, we added this information

P 8 l 28: The term "jerky movement" requires a reference (Goldthwait ?). This is another somewhat speculative sentence which is ok in a discussion but misplaced in a conclusion.

Sentence deleted

P 8 l 31: "As conventional…" is this sentence complete? After reading it multiple times I think I understand that "ELA" and "specific mass balance" are other glaciological parameters to characterize a glaciers, but you find them less well suited compared to the ice-flow velocity.

We added: "… long-term monitoring of ice flow provides valuable additional information on the glacier state."

P 8 l 30 "extra effort" compared to what?

deleted

I apologize, but the meaning of the last paragraph is opaque to me. Please rephrase. I suggest ending the paper with your main finding, uniqueness and further applications of your dataset. Keep it succinct.

changed

[revised manuscript text omitted]